# Thoracic Sympathectomy for Primary Hyperhidrosis: Focus on Post-Operative Age-Related Quality of Life

**Gaetano Romano** [1,*], **Federico Davini** [1], **Alessandra Lenzini** [2], **Carmelina Cristina Zirafa** [1] and **Franca Melfi** [1]

1 Minimally Invasive and Robotic Thoracic Surgery, University Hospital of Pisa, 56124 Pisa, Italy
2 Department of Surgical, Medical, Molecular and Critical Care Pathology, University of Pisa, 52124 Pisa, Italy
* Correspondence: gaetano.romano@ao-pisa.toscana.it; Tel.: +39-349-823-5342; Fax: +39-050995218

**Abstract:** Primary focal hyperhidrosis is an idiopathic condition characterized by excessive sweating, predominantly localized in the hands and armpits. This condition affects about 1% of the general population and it is often associated with a deterioration of the Quality of Life (QoL), especially in younger patients. Medical therapy, usually prescribed as a first approach, is associated with good results, but only in the short term. Surgery, on the other hand, is associated with a definitive resolution of the disease in most patients. Currently, there is no consensus on the timing of treatment and the final decision is often at the discretion of the physician and the patient. The aim of this study is to analyze the post-operative, age-related QoL in patients affected by primary hyperhidrosis treated by surgery by analyzing data of 56 patients who underwent biportal thoracoscopic sympathectomy between January 2016 and October 2019, dividing the patients into two groups: under and over the age of 25. The QoL was studied by administering the IIRS questionnaire pre-operatively and then six months after surgery. Data analysis demonstrated a lower complication rate in younger patients and equal post-operative outcomes in the two age groups.

**Keywords:** sympathectomy; hyperhidrosis; thoracoscopy; quality of life; thoracic surgery; minimally invasive surgery

## 1. Introduction

Primary hyperhidrosis (PH) is defined as a pathologic condition related to hypersecretion of sweat glands, associated with sympathetic hyperactivity [1]. This condition is not a direct consequence of temperature or emotional stimuli, but it depends on an idiopathic dysfunction of sympathetic nerve transmission [2]. PH affects nearly 1% of the population, involving all age groups, although the majority of patients are adolescents or young adults [3]. Severe hyperhidrosis is predominantly localized in the axillae, palms, face, and feet, with important impairment on Quality of Life (QoL), as documented in several studies. Patient often suffer extreme social anxiety which may result in the development of major social, scholar, and professional discomfort [4]. Indeed, Hasimoto and colleagues, in 2018, have argued that psychological distress and emotional embarrassment in PH patients can be considered similar to those caused by chronic diseases such as psoriasis and kidney failure [5]. Currently, different guidelines for the treatment of PH are available [6,7], and consequently, the appropriate approach is established by the clinicians depending on the case [8].

There are a variety of treatments available for PH based on the use of topical agents, systemic therapies, and surgery. Topical aluminum-based antiperspirants, iontophoresis, and botulin toxin injection are considered as the first and second-line treatment in focal PH, depending on the severity of the case [9]. Moreover, for non-responsive patients, some authors recommend the use of oral anticholinergics as oxybutynin and glycopyrrolate [10]. However, although these medical treatments alleviate the symptoms and are surely less invasive than a surgical treatment, their effect can be only temporary and can constitute burdensome spending for the patients [11]. Conversely, to date, surgical therapy can be

considered the most resolutive treatment in PH patients, although some authors state that it should be reserved as a "last-chance" option, being used after a proven failure of first and second-line non-surgical therapies [8]. Minimally invasive (endoscopic) thoracic sympathectomy (ETS) is considered the gold standard treatment for axillary and palm hyperhidrosis [12]. Both uniportal and biportal approaches have been demonstrated to be safe, effective, and standardized procedures. Furthermore, several studies have demonstrated that ETS is also associated with a good aesthetic result, short post-operative stay, little post-operative pain, and Quality of Life (QoL) improvement in PH patients [11]. Nevertheless, ETS can be associated with some undesirable side effects, such as compensatory sweating (CS), which is considered the most common mid-term complication. CS can occur predominantly in the trochus, abdomen, and lumbar region in patients in whom ETS is proven to be effective to solve excessive sweating in the palms and axillae. Some authors have shown that this secondary effect can be verified at a rate ranging between 3% and 98% of patients [13], though, at present, there is no literature consensus between CS and different surgical techniques. Analyzing the scientific evidence, as previously exposed, the debate on the pattern of treatment and the age-related choice of PH therapy is still open, especially the timing of the surgical indication. Consequently, univocal comparative data concerning the QoL related to the age of surgery are not available. In view of the foregoing, the aim of this study is the assessment of QoL outcomes in PH patients who underwent bilateral ETS, also comparing the result obtained in the two-age group analyzed.

## 2. Materials and Methods

A retrospective chart review of all patients who underwent ETS for PH between January 2016 and October 2019 in the Division of Minimally Invasive and Robotic Thoracic Surgery of Pisa was conducted.

### 2.1. Patients

Fifty-six consecutive patients were included; 25 females (44.6%) and 31 males (55.4%), with a mean age of 30.5 years (SD 9.66) underwent 112 thoracoscopic sympathectomy under general anesthesia. In preparation for surgery, blood exams, physical examination, and anesthesiologic evaluation were performed. A chest X-ray was executed pre-operatively to exclude lung diseases and a cardiological examination was requested to detect possible rhythm dysfunctions. All patients with possible secondary hyperhidrosis were excluded from our cohort. In fact, all patients underwent a dosage of thyroid hormones, in order to exclude hyperthyroidism, and underwent endocrinological evaluation, if required. Patients were divided into two groups: Group A, patients under the age of 25 and Group B, over the age of 25. The subdivision into two groups was performed in order to observe any different surgical outcomes related to the different age of the patients (Table S1).

### 2.2. Surgical Technique

Each patient underwent bilateral ETS within the same operating session. The sympathectomy was performed placing the patient in a lateral decubitus position, with the body flexed ventrally and the ipsilateral abducted arm, mimicking the swimming "freestyle stroke". Two centimetric thoracoscopic accesses were executed on each side at the II (or III) and the IV (or V) intercostal space. Once the surgery was performed on one side (right or left, interchangeably), the patient was placed in the opposite decubitus to perform the contralateral procedure. The sympathectomy was performed bilaterally using a 30°–5 mm camera and a 5 mm endoscopic Hook, cauterizing the sympathetic chain from T3 to T4, taking care to dissect any collateral nerve fibers (Kuntz nerves). Once the dissection of the nerve and the collateral fibers was completed, these were removed and sent for histological examination. At the end of the procedure, a small 20 Fr tube was placed on each side through the lower access to ensure a complete lung expansion (Figure 1) and a post-operative chest X-ray was obtained for all patients 3–4 h after surgery.

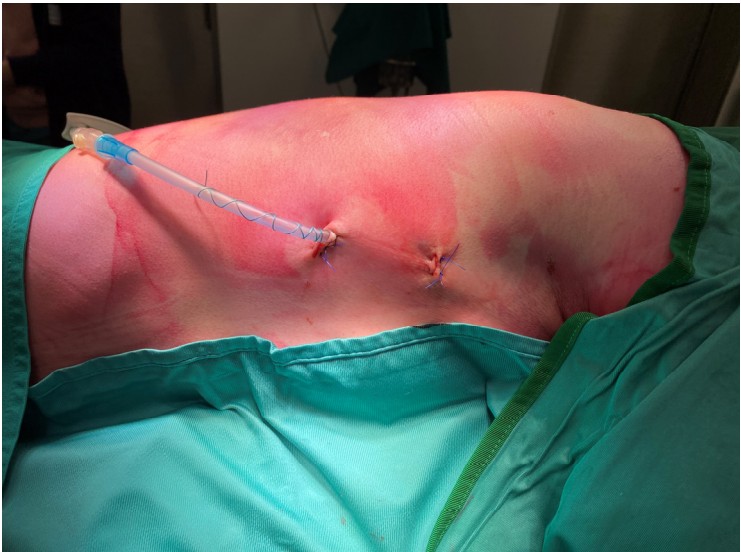

**Figure 1.** Post-operative picture showing surgical ports after chest tube placement.

*2.3. Anesthesiological Protocol*

All surgical interventions were executed under general anaesthesia and selective oro-tracheal intubation, according to the following pharmacological plan:

- Propofol 2%, 5 mg/kg/hour
- Ramifentanil 0.2 mcg/kg
- Morphine 0.1 mg/kg intramuscularly at induction of anesthesia
- Paracetamol 1 g
- Ketorolac 30 mg
- Rocuronium 0.6 mg/kg

Post-operative pain was contained with the administration of acetaminophen 1 g every 8 h on the first post-operative day and subsequently only in case of pain.

*2.4. QoL Assessment*

The evaluation of Quality of Life (QoL) parameters was performed by IIRS (Illness Intrusiveness Rating Scale) questionnaire administration (Table 1), conducting the interview by telephone or during the outpatient check-up by one of the authors. The questionnaire was administered pre-operatively and then 6 months after surgery. The IIRS questionnaire evaluates 10 different items through 10 questions concerning: any previous treatment, discomfort in personal life, any disability in hygiene habits, and emotional distress in relationships and recreational life. The total compilation of the questionnaire allows to obtain a score from 0 to 70, directly proportional to the degree of discomfort of the patient. Patients were also asked to attribute a numerical value to the discomfort perceived due to post-operative compensatory sweating and the effectiveness of surgery with a score from 0 to 7.

During the study, no patient refused to participate in the survey. None of the patients were lost to follow up.

**Table 1.** IIRS (Illness Intrusiveness Rating Scale) questionnaire administration.

| 1 | How much sweating affect your daily life? | 0–7 |
|---|---|---|
| 2 | Do you use cosmetics to reduce sweating? How many times a day? | 0–7 |
| 3 | Do you use drugs to reduce sweating? How many times a day? | 0–7 |
| 4 | Do you use physical treatments or electrical devices to reduce sweating? How many times a day? | 0–7 |
| 5 | Does sweating force you to take multiple showers a day? How many times? | 0–7 |
| 6 | Did sweating force you to change your clothes several times a day? | 0–7 |
| 7 | Does excessive sweating limit the choice of clothes in terms of color and material? | 0–7 |
| 8 | Do alcoholic or caffeinated drink increase sweating? | 0–7 |
| 9 | Does eating spicy food increase sweating? | 0–7 |
| 10 | Does being in a state of psychological stress increase sweating? | 0–7 |

*2.5. Statistical Analysis*

For the statistical data evaluation, the Excel-365 and XL-STAT program for Mac (Version 16.36) was used. Continuous variables were analyzed using the T-student test. The *p*-value resulting from the comparison of the average indexes was considered statistically significant if less than 0.05. This study was approved by our Institutional Review Board. Each eligible patient voluntarily signed an informed consent.

**3. Results**

In this study, 56 consecutive patients who underwent 112 ETS were evaluated. The series was divided into two groups according to the age of the patients: Group A (under 25 years old) included 23 patients: 11 females (47.8%) and 12 males (52.2%), the mean age resulted in 20.8 years (SD = 2.5) with an age range of 16–24 years. In this group, two patients were affected by axillae hyperhidrosis (8.6%), four palm hyperhidrosis (15.4%), and 17 were suffering from both axillary and palm hyperhidrosis (76%). Group B (over 25 years old) included 33 patients: 14 females (42.4%) and 19 males (57.6%), the mean age resulted in 37.3 years (SD = 6.3) with a range of 28–51 years. In this group, five patients were affected by axillae hyperhidrosis (15.1%), 11 palm hyperhidrosis (33.3%), and 17 were suffering from both axillary and palm hyperhidrosis (51.6%). The surgery was performed bilaterally for each patient within the same operating session. No intra-operative complications occurred. No surgery required conversion to thoracotomy. The mean operating time was 62 min, including the patient's decubitus change. All patients experienced a resolution of the hyper-perspiration immediately after surgery: they referred that their hands and armpits appeared warm and dry after recovery from general anaesthesia. All patients underwent a post-operative chest X-ray; pleural drainages were removed on the first day after surgery. Further, 55/56 patients (98.2%) were discharged 24 h after surgery. One patient (1.8%) in Group B was transferred to a pneumology unit for developing post-operative pneumonia. No cases of pneumothorax occurred, and no patient referred Bernard Horner Syndrome. One patient in Group B reported the onset of dysesthesia in the left hemithorax six months after surgery. The mean pre-operative IIRS scores resulted in 19.4 (SD 5.3) and 17.8 (SD 5.9) for group A and B, respectively. The mean post-operative score for group A was 3.2 (SD 2.7), while the IIRS mean score resulted in 4.2 (SD 4.4) for Group B. The statistical analysis showed that the difference between the average of the pre- and post-operative scores is statistically significant for both groups (*p* < 0.01). No statistically significant difference was found comparing the mean IIRS post-operative score between the two groups. Based on the responses to the IIRS questionnaire, 14 patients (25%) complained of impairment due to compensatory sweating: six (26%) in group A, eight (24%) in group B, attributing to this phenomenon a score greater than or equal to 5/7. However, 52 patients (92.8 %) declared their satisfaction with the effectiveness of surgery, given a score equal to or greater than 6/7 six months after surgery.

## 4. Discussion

Primary hyperhidrosis affects 1% of the general population and, particularly, 1.6% of adolescents and young adults. Several studies have demonstrated that PH has a strongly negative impact on different aspects of life: relationship with the partner, friendships, family life, scholar, and work activities (such as difficulties in using a pencil or a pen, keeping sheets dry, manipulating a computer mouse or keyboard) [14]. Nowadays, multiple medical treatments are available. Aluminum-based antiperspirants, iontophoresis, botulin toxin injection, and oral anticholinergics agents are widely used in the treatment of PH. Nonetheless, these non-surgical therapies possess some not negligible side effects and time-limited results. In view of this, frequently, the discomfort complained by the patients becomes so oppressive, as to motivate them to go to the surgeon for the purpose of solving the embarrassment and discomfort caused by PH [1]. Contrary to dermatological agents, surgical therapy (thoracoscopic sympathectomy: ETS) has proven to be safe, repeatable, and effective in resolving the symptoms. In effect, several reports have established a 95% rate of success in the resolution of palmar hyperhidrosis [15]. Before the advent of thoracoscopic surgery, in the late 1980s, thoracotomy was the only possible approach for the treatment of PH, with hard consequences for the patient related to hospitalization, complications, and post-operative pain [16]. The minimally invasive surgery, conversely, has represented a real revolution in PH treatment. VATS is perfectly fit for benign disease: determining a better aesthetic result, shorter hospitalization, lower rate of intraoperative and post-operative complications, and the same efficacy as traditional surgery [17]. Currently, the rationale for surgery consists of the ablation of the sympathetic chain from T2–T3 to T4–T5 in order to block the nerve conduction involved in the hypersecretory stimulus. Although the level of ablation still remains controversial, numerous studies have shown that ablation starting from T3 is associated with better efficacy and lower post-operative complication, including hand and trunk compensatory sweating. In our center, the elective surgery for the treatment of PH is the ETS by biportal approach with the ablation of T3–T4 tract of the sympathetic chain. The interruption is obtained by isolating, cutting, and removing the nerve, including the ganglion and Kuntz fibers. In this series, our technique showed encouraging results in terms of the resolution of symptoms with a low post-operative complication rate (1.8%). This result is significant when compared with the data in the literature, in fact, the reported incidence of complications for ETS can reach 10% [18]. The fact that no pneumothorax occurred in our series is probably linked to the positioning of two pleural drainages (one per side) maintained for 24 h and removed only after radiological confirmation. The two complications reported (one pneumonia and one post-operative dysesthesia) occurred in the over-25 group, while no patient in Group A had post-operative complications. This observation could further justify the earliness of intervention in younger patients, who are more suitable for surgery, since they can tolerate the physiological insult posed by surgical procedure, with low peri-operative risk. In addition, early treatment of hyperhidrosis, even if with an invasive technique, could avoid years of discomfort in younger patients. The rate of compensatory sweating presented in this study was 25%: 14 patients: 6 (26%) in Group A and eight (24%) in Group B. Even if it is well established that compensatory sweating is the most common side effect following sympathectomy, the reported incidence of this complication varies greatly in scientific literature, with an incidence ranging from 10 to 90% [19,20]. The wide variability may be attributable to heterogeneous patient populations and a variety of surgical techniques. Moreover, the measuring method used to evaluate compensatory sweating is not univocal and not yet standardized. The analysis of the QoL scores documented a statistically significant difference in pre- and post-operative values in both groups. The pre-operative IIRS indicates an important value of discomfort, which significantly decreased in the post-operative period. This result, in accordance with the data in the literature, further demonstrates the effectiveness of the surgical treatment of PH. Moreover, we evaluated the difference in the post-operative IIRS scores between the two groups, obtaining a not statistically significant result. Thus, we can state that the

post-operative QoL outcomes can be considered similar, hence recommending the surgical indication also in younger patients.

## 5. Conclusions

Primary focal hyperhidrosis is a condition that can seriously deteriorate the Quality of Life of a patient. To date, the therapeutic approach and the appropriate timing for the intervention still remain widely debated. Currently, data on the QoL in relation to the age of the patients with PH undergoing ETS are not available in the literature. The results obtained from this retrospective study document a lower rate of post-operative complications in younger patients and an equal efficacy of surgery compared to adult patients.

Hence, our data in addition to being consistent with the literature from a general point of view [21] could furthermore justify the use of surgery even in younger patients. In fact, the post-operative outcomes of this series show that surgery can be proposed and performed in young and motivated patients with PH. However, studies with long-term results and with a larger number of patients will be required.

**Supplementary Materials:** The following supporting information can be downloaded at: https://www.mdpi.com/article/10.3390/surgeries4010014/s1, Table S1. Sample characteristics and results.

**Author Contributions:** Conceptualization, G.R.; methodology, G.R.; validation, F.M. and C.C.Z.; formal analysis G.R.; investigation, G.R. and F.D.; resources, F.D. and A.L.; data curation, A.L.; writing—original draft preparation, G.R. and C.C.Z.; writing—review and editing, G.R. and C.C.Z.; visualization, G.R.; supervision, F.M.; project administration, G.R. and F.M. All authors have read and agreed to the published version of the manuscript.

**Funding:** This research received no external funding.

**Institutional Review Board Statement:** Not applicable.

**Informed Consent Statement:** Informed consent was obtained from all subjects involved in the study.

**Data Availability Statement:** Data are contained within the article.

**Acknowledgments:** Thanks to Teresa Hung Key for proofreading.

**Conflicts of Interest:** The authors declare no conflict of interest.

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
