# Peer review of "Thoracic Sympathectomy for Primary Hyperhidrosis: Focus on Post-Operative Age-Related Quality of Life"

_2673-4095, doi:10.3390/surgeries4010014_

Round 1

Reviewer 1 Report

Dear authors,

Thank you for the opportunity to participate in reviewing your paper, which I think is quite interesting.

Although there are no precise guidelines on the selection of candidates for surgery, the operative treatment of hyperhidrosis has a not neglectable impact on quality of life. Without solid indications or transparent consensus papers, surgical treatment at high-volume centers and with a minimally invasive technique should be provided.

I suggest only a minor language review and spell check.

Congratulations on your work.

Author Response

Thank you for the opportunity to participate in reviewing your paper, which I think is quite interesting.

Although there are no precise guidelines on the selection of candidates for surgery, the operative treatment of hyperhidrosis has a not neglectable impact on quality of life. Without solid indications or transparent consensus papers, surgical treatment at high-volume centers and with a minimally invasive technique should be provided.

I suggest only a minor language review and spell check

Thank you for your valuable opinion and for your favorable opinion on our study.
We revised the spelling of the article to correct any errors

Reviewer 2 Report

Thank you for the opportunity to review this manuscript, which reports the results of a series of patients undergoing bilateral thoracoscopic thoracic sympathectomy for primary hyperidrosis.

Strengths

- The topic is of interest, and I believe the use of a Quality of Life assessment to be even more relevant when investigating this particular disease.

Weaknesses

- A higher attention to details could improve the quality of the article (typos, syntax mistakes, minor spell checks (Line 15, line 36, line 156, etc.))

- Some statements in the text are not supported by a reference; moreover, I think the authors could make the manuscript more interesting by discussing more thoroughly the impact of compensatory sweating after surgery

- The use of tables reporting the data could make the results easier to understand.

I have some comments:

Introduction

-In line 40, the authors say there are no revised guidelines by the FDA, thus implying the existence of previously published but not revised recommendations by the FDA: if that is the case, they should be included in the references; otherwise, I think readers might find this statement, and the references following it, unclear.

-In line 59 I think there is a typo: “Age Related, post-operative QoL in Sympathectomy Nevertheless, ETS can be associated with some undesirable side effects[…], could the authors please check this sentence?

- Line 64“Some authors have shown that this secondary effect can be verified at a rate ranging between 3% and 98% of patients, though, at present, there is no literature consensus between CS and different surgical techniques”: I suggest the authors add at least one reference reporting these prevalence rates.

Methods

-During the study period did any patient refuse to participate to the study? Was any patient lost at follow-up? Who administered the QoL questionnaire?

- Based on what criterion did the authors decide to divide the patients in two Groups using 25 years of age as cut-off?

-Did the authors also analyse the difference between pre- and postoperative IRSS score for each patient and not based on the mean IRSS score of the group?

-I think it would be interesting to include data regarding medical treatment previously received by the patients enrolled in the study

Results

-The Groups are defined as A and B and then as Group II. Could the authors fix this?

-I suggest the authors include a table to depict demographic characteristics of the two groups, preoperative symptoms and pre and postoperative IRSS scores.

-Table 1: questions 6 and 7 are the same

Discussion

-Line 195: “This result is significant when compared with the data in the literature, in fact, the reported incidence of complication for ETS can reach 10%, reaching up to 30% when considering only pneumothorax.” Authors should report a reference supporting these data

-Line 200: “This observation could further justify the earliness of intervention in younger patients, who are more suitable for surgery.”: could the authors explain in which sense they believe younger patients are more suitable for surgery? I am not sure this concepts is clearly described in the manuscript

-The authors do not discuss the IRSS preoperative score, which is 19.4 over 70 and 17.8 over 70 in the two groups, respectively. I believe this could be of interest.

-Could the authors discuss their rate of compensatory sweating, with reference to the existing literature?

Conclusions

-Line 219 “In fact, the postoperative outcomes of this series show that surgery can be proposed and performed in young and motivated patients with PH, without resorting to preliminary medical  therapies.”: I believe this statement should be supported by a comparison with the effectiveness of medical therapies, which is not the aim of this study

References

The authors may consider including the STS guidelines on hyperidrosis published in 2011 (Cerfolio et al. Ann Thorac Surg 2011;91:1642–8).

Author Response

- A higher attention to details could improve the quality of the article (typos, syntax mistakes, minor spell checks (Line 15, line 36, line 156, etc.)

Thank you for the suggestion we corrected the minor errors

- Some statements in the text are not supported by a reference; moreover, I think the authors could make the manuscript more interesting by discussing more thoroughly the impact of compensatory sweating after surgery

Thank you for the suggestion, we have improved the text focusing on more literature data.

- The use of tables reporting the data could make the results easier to understand.

 Thank you for the suggestion, we added a summary table

I have some comments:

Introduction

-In line 40, the authors say there are no revised guidelines by the FDA, thus implying the existence of previously published but not revised recommendations by the FDA: if that is the case, they should be included in the references; otherwise, I think readers might find this statement, and the references following it, unclear.

the sentence has been modified in order to avoid ambiguity

-In line 59 I think there is a typo: “Age Related, post-operative QoL in Sympathectomy Nevertheless, ETS can be associated with some undesirable side effects[…], could the authors please check this sentence?

Sentence corrected as you suggested

- Line 64“Some authors have shown that this secondary effect can be verified at a rate ranging between 3% and 98% of patients, though, at present, there is no literature consensus between CS and different surgical techniques”: I suggest the authors add at least one reference reporting these prevalence rates.

Reference added in the text and in the bibliography as you suggested

Methods

-During the study period did any patient refuse to participate to the study? Was any patient lost at follow-up? Who administered the QoL questionnaire?

The questionnaire was administered by telephone interview or outpatient check up by one of the authors (as added in the text in line 117).

The other information required in this comment are exposed in line 126-127

- Based on what criterion did the authors decide to divide the patients in two Groups using 25 years of age as cut-off?

The criterion on which we performed our study was precisely age.

The authors wanted to observe if age itself could influence surgical outcomes. Authors chose the age of 25 because it represents the average age at which many patients undergo surgery for this disease. (info addend in the text in line 88-89)

-Did the authors also analyse the difference between pre- and postoperative IRSS score for each patient and not based on the mean IRSS score of the group?

We analyzed the differences in pre- and post-operative scores for each patient. The mean value reported in the text is the result of the sum of the patients' scores, divided by the number of patients themselves.

-I think it would be interesting to include data regarding medical treatment previously received by the patients enrolled in the study

Thanks for the kind suggestion, this information is certainly interesting but it was not investigated in our study. However, it will certainly be the focus of future experiences.

Results

-The Groups are defined as A and B and then as Group II. Could the authors fix this?

Thanks, we have corrected the text as suggested.

-I suggest the authors include a table to depict demographic characteristics of the two groups, preoperative symptoms and pre and postoperative IRSS scores.

Tab 2 added as you suggested

-Table 1: questions 6 and 7 are the same

Thank you, the correct n.6 question is: “Did sweating force you to change your clothes several times a day?”. We have corrected the table

Discussion

-Line 195: “This result is significant when compared with the data in the literature, in fact, the reported incidence of complication for ETS can reach 10%, reaching up to 30% when considering only pneumothorax.” Authors should report a reference supporting these data

-Thanks for the suggestion, we added the reference number 19 and modified the text to avoid ambiguity.

Pedro M. Rodríguez, Jorge L. Freixinet, Mohamed Hussein, Jose M. Valencia, Rita M. Gil, Jorge Herrero, Araceli Caballero-Hidalgo, Side effects, complications and outcome of thoracoscopic sympathectomy for palmar and axillary hyperhidrosis in 406 patients, European Journal of Cardio-Thoracic Surgery, Volume 34, Issue 3, September 2008, Pages 514–519, https://doi.org/10.1016/j.ejcts.2008.05.036

-Line 200: “This observation could further justify the earliness of intervention in younger patients, who are more suitable for surgery.”: could the authors explain in which sense they believe younger patients are more suitable for surgery? I am not sure this concepts is clearly described in the manuscript.

Young patients are usually the more “fit for surgery”, since they can tolerate the physiological insult posed by surgical procedure, with a low peri-operative risk. In addition, early treatment of hyperhidrosis, even if with an invasive technique, could save young patients from many years of discomfort. However, the determination of  the optimal timing of surgical treatment goes beyond the objective of our analysis. Ws added this information in line 208-210

-The authors do not discuss the IRSS preoperative score, which is 19.4 over 70 and 17.8 over 70 in the two groups, respectively. I believe this could be of interest.

The mean pre-operative IRSS scores resulted in 19.4 (SD 5.3) and 17.8 (SD 5.9) for group A and B respectively (line 161). The score obtained is proportional to the degree of discomfort and impairment of patients, related to hyperhidrosis, and estimates the psychosocial impact of the disorder as well as the effectiveness of treatment. We added this information in line 212-214

-Could the authors discuss their rate of compensatory sweating, with reference to the existing literature?

Our rate of compensatory sweating was 25%: 14 patients: 6 (26%) in group A and8 (24%) in group B Lines 167-167). Even if it is well established that compensatory sweating is the most common side effect following sympathectomy, the reported incidence of this complication varies greatly in scientific literature, with an incidence ranging from 10 to to 90%. The wide variability may be attributable to heterogeneous patient populations and a variety of surgical techniques. Moreover, the measuring method used to evaluate compensatory sweating is not univocal and not yet standardized.

  • Wolosker N, Milanez de Campos JR, Fukuda JM. Management of Compensatory Sweating After Sympathetic Surgery. Thorac Surg Clin. 2016 Nov;26(4):445-451.
  • Deng B, Tan QY, Jiang YG, Zhao YP, Zhou JH, Ma Z, et al. Optimization of sympathectomy to treat palmar hyperhidrosis: the systematic review and meta-analysis of studies published during the past decade. Surg Endosc. 2011;25(6):1893–901.

We added the references and the discussion in lines 213-219

Conclusions

-Line 219 “In fact, the postoperative outcomes of this series show that surgery can be proposed and performed in young and motivated patients with PH, without resorting to preliminary medical  therapies.”: I believe this statement should be supported by a comparison with the effectiveness of medical therapies, which is not the aim of this study.

Modified as you suggested to avoid ambiguity.

References

The authors may consider including the STS guidelines on hyperidrosis published in 2011 (Cerfolio et al. Ann Thorac Surg 2011;91:1642

Added ref number 22.

Reviewer 3 Report

Dear Author, the study is well designed and the paper is well written. It should be interesting to know the operative time and to see an operative picture. 

Author Response

Dear Author, the study is well designed and the paper is well written. It should be interesting to know the operative time and to see an operative picture. 

Thank you for your valuable suggestions. We added the informations you required in the text. (mean operating time  and operative picture).